# Repeat induces not only gene silencing, but also gene activation in mammalian cells

**Yusuke Ogaki, Miki Fukuma, Noriaki Shimizu** *

Graduate School of Biosphere Science, Hiroshima University, Higashi-Hiroshima, Hiroshima, Japan

* shimizu@hiroshima-u.ac.jp

## Abstract

Repeat-induced gene silencing (RIGS) establishes the centromere structure, prevents the spread of transposons and silences transgenes, thereby limiting recombinant protein production. We previously isolated a sequence (B-3-31) that alleviates RIGS from the human genome. Here, we developed an assay system for evaluating the influence of repeat sequences on gene expression, based on *in vitro* ligation followed by our original gene amplification technology in animal cells. Using this assay, we found that the repeat of B-3-31, three core sequences of replication initiation regions (G5, C12, and D8) and two matrix attachment regions (AR1 and 32–3), activated the co-amplified plasmid-encoded *d2EGFP* gene in both human and hamster cell lines. This upregulation effect persisted for up to 82 days, which was confirmed to be repeat-induced, and was thus designated as a repeat-induced gene activation (RIGA). In clear contrast, the repeat of three bacterial sequences (lambda-phage, Amp, and ColE1) and three human retroposon sequences (Alu, 5'-untranslated region, and ORF1 of a long interspersed nuclear element) suppressed gene expression, thus reflecting RIGS. RIGS was CpG-independent. We suggest that RIGA might be associated with replication initiation. The discovery of RIGS and RIGA has implications for the repeat in mammalian genome, as well as practical value in recombinant production.

## Introduction

Amplification of oncogenes or drug-resistant genes plays a pivotal role in malignant transformation of human cells. Amplified genes frequently localize at the extrachromosomal double minutes (DMs) or the chromosomal homogeneously staining region (HSR). We have previously reported that a plasmid bearing both a replication initiation region (IR) and a nuclear matrix attachment region (MAR) efficiently mimics gene amplification, and spontaneously generates DMs and HSR in transfected cells [1, 2]. Since this discovery, we have been using this IR/MAR gene amplification technology to uncover the underlying mechanism of gene amplification [3–5], investigate basic cell biology (see [6] for an early review), and efficiently produce recombinant proteins [7–9]. However, for recombinant proteins production, a major drawback was that protein production did not necessarily increase in proportion to the increase in gene copy number [10], since the amplified genes frequently formed a silent chromatin [11, 12]. This was caused by IR/MAR plasmid amplification into the tandem repeat

**Funding:** Japan Society for the Promotion of Science, Grant/Award Number: a Grant-in-Aid for

Scientific Research (C) (16K081) to N.S. in part. The funders had no role in study design, data collection and analysis, decision to publish, or preparation of the manuscript. This research was supported by AMED under Grant Number JP19ae0101054

**Competing interests:** The authors have declared that no competing interests exist.

structure [2], which was easily silenced by a mechanism known as repeat-induced gene silencing (RIGS) (for reviews see [13, 14]).

RIGS has an important cellular function by heterochromatinizing repeated sequences, such as those in the pericentric region to increase mechanical strength [15], ultimately silencing the repeated transposon sequence to prevent its spread [16] with a similar effect to transfected genes [17, 18]. Furthermore, 250 to 670 copies of ribosomal DNA were tandemly repeated in diploid human genome, however only a fraction of them is transcriptionally active [19]. The involvement of an RNA interference mechanism was proposed to silence the tandem repeat [20–22]. RNA interference was also suggested for nucleolar organization and repeated DNA stability [23] or pericentric heterochromatin formation [24]. Involvement of RNA interference in the latter case was bypassed by the elimination of histone H3K14 acetyl transferase [15]. Heterochromatin assembly in transgene repeats is independent of RNA interference [25]}. It has recently been reported that a complex bearing retinoblastoma protein extensively occupies and represses expression of genomic tandem repeat [26].

Our previous investigation of the epigenetic chromatin status of amplified sequences at the DMs or HSRs, generated using our IR/MAR technology, led us to propose the "DNA methylation-core and heterochromatin-spread" model for RIGS [12]. To alleviate RIGS, we screened the human genomic library and obtained a 3,271-bp sequence, named "B-3-31", that enhanced gene expression from tandemly amplified repeats [27]. Further, we determined the core (minimum) sequences that supported gene amplification inside the c-*myc* IR (2349 bp), *DHFR* IR (*Oriβ*; 4634 bp), and β-globin Rep-P IR (2771 bp), and obtained the sequences "C12" (798 bp) [28], "D8" (2347 bp) [28], and "G5" (972 bp) [29], respectively. Importantly, when we ligated G5 DNA to the direct or inverted repeat *in vitro*, the G5 repeat DNA was efficiently amplified in the transfected cells, similar to the IR/MAR plasmid amplification [30]. Furthermore, the co-transfected plasmid was co-amplified in the long stretch of amplified G5, and expression from the plasmid-encoded genes was significantly upregulated [30]. Although this increase in gene expression might be expected since G5 is a core IR, and IRs were reported to upregulate neighboring genes [31], the fact that repeated G5 was transfected and then further amplified in cells suggested that the repeat *per se* does not necessarily result in RIGS, and that another type of repeat might instead efficiently upregulate the neighboring gene. We named this hypothetical phenomenon as "repeat-induced gene activation (RIGA)" as a contrast to the RIGS phenomenon. Thus, the aim of the present study was to verify the existence of sequence-dependent RIGA.

## Materials and methods

### Experimental design to evaluate RIGS or RIGA

The overall strategy is outlined in Fig 1A. We have previously found that any DNA, including lambda-phage DNA, might be co-amplified in the cells when co-transfected with an IR/MAR plasmid [2]. Subsequently, we conducted a series of experiments to exploit this phenomenon for amplifying various sequences in cells (e.g., [3, 9, 32, 33]. The co-transfected DNA was recombined in the cells, and was further multimerized to a large circular molecule, in which the transfected sequences were arranged as direct repeats [2]. This DNA could frequently recombine with extrachromosomal DMs if present in the same cells [3]. Such extrachromosomal DNA of tandem repeats might then be integrated to the chromosome arm and further amplified by inducing the breakage-fusion-bridge cycle to generate HSR [3]. Based on these previous findings, we used this system to evaluate whether the repeat of a test sequence may downregulate or upregulate the expression of the gene of interest, destabilized enhanced green fluorescent protein (*d2EGFP*) in this case, among the amplified tandem array of transfected

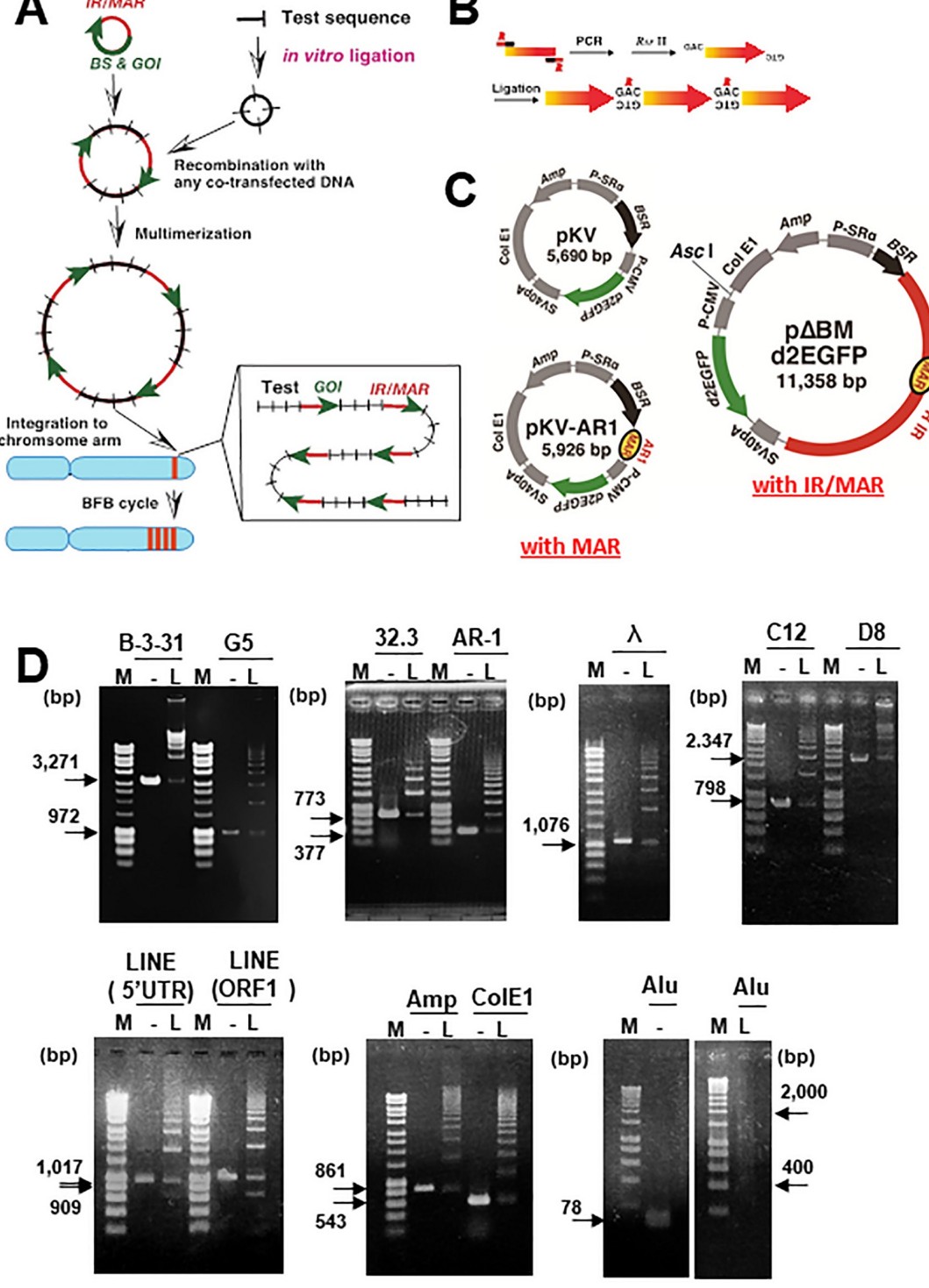

**Fig 1. Overview of the experimental set-up to evaluate RIGS ad RIGA.** (A) Schematic of the experimental strategy. (B) Preparation of a direct-repeat DNA sequence using PCR amplification, *Rsr*II digestion, and ligation. (C) Structure of the plasmids used in this study. (D) Electrophoretic analyses of the ligation products introduced to the cells. M; molecular weight marker, -; unligated DNA, L; ligated DNA.

sequences generated in cells. briefly, the test sequence was amplified using polymerase chain reaction (PCR), digested by *Rsr*II at the PCR primer, and ligated *in vitro*. Since *Rsr*II recognizes a non-palindromic sequence, the ligation produces only the direct repeat of the test sequence. The repeat DNA was then mixed with the DNA of an IR/MAR-bearing plasmid (pΔBM d2EGFP) in most experiments, and co-transfected in human and hamster cells, respectively, as detailed below.

### Preparation of repeat DNA

The following sequences were PCR-amplified using the primers listed in S1 Fig. Human genomic B-3-31 (3271 bp) [27] was cloned from a human genomic library, which was previously shown to alleviate RIGS as demonstrated by the amplified IR/MAR plasmid. B-3-31 was originally cloned at the *Asc*I site of the IR/MAR-bearing plasmid pΔBM d2EGFP. AR1 (377 bp) is an intronic MAR from the mouse Igκ gene [34], which we have frequently used for IR/MAR amplification in previous studies. Sequence 32–3 (773 bp) is an unpublished sequence, which was a kindly gifted by Dr. Ken Tsutsui (Okayama University). This sequence was originally cloned from rat genomic library and showed strong *in vitro* MAR activity (Ken Tsutsui, personal communication). The sequences G5 (971 bp [29]), C12 (798 bp [28]), and D8 (2347 bp [28]) are core IRs from the β-globin, c-*myc*, and *DHFR* genes, respectively, and were isolated as the shortest sequences capable of supporting gene amplification. The lambda-phage sequence used in this study was derived from position 1891–2966 (1076 bp) of lambda-phage (NCBI NC_001416.1). The ampicillin-resistance gene sequence was derived from position 6171–7031 (861 bp) of pCEP4 (10,186 bp; https://www.snapgene.com/resources/plasmid-files/?set=mammalian_expression_vectors&plasmid=pCEP4), and the Colicin E1 (ColE1) sequence was derived from position 2594–3136 (543 bp; ORF frame 1 in https://www.addgene.org/browse/sequence_vdb/1433/). The Alu sequence was derived from position 52–129 (78 bp) of the consensus sequence appearing in Weisenberger et al. [35]. The long interspersed nuclear element (LINE) sequence was obtained from https://www.ebi.ac.uk/ena/data/view/AH005269, and the 5′ untranslated region (UTR; position 306–1214, 909 bp) and ORF1 (position 1216–2232; 1017 bp) were amplified using human genomic DNA as a template.

Each PCR primer had a 20-nt sequence specific to the target, which was followed by an 11–14-nt sequence with a *Rsr*II recognition site at the 5′ end. *Rsr* II cuts non-palindromic sequences, which can only be ligated as a direct repeat (Fig 1B). Following PCR amplification using these primer sets, KOD-Plus Neo DNA polymerase (Toyobo Co.), and template DNA, the products were digested by *Rsr*II (New England Biolabs Inc.). The DNA was purified using NucleoSpin1 Gel and PCR Clean-up kits, according to the manufacturer instructions, (MACHEREY-NAGEL Co.) and was ligated using Ligation high Ver.2 (TOYOBO Co.). The ligated DNA was purified by phenol/chloroform extraction and ethanol precipitation, and finally used for cell transfection. The electrophoretic analysis of the preparation is shown in Fig 1D.

### Plasmids

The repeat DNA was mixed with the plasmids pKV, pKV-AR1, and pΔBM.d2EGFP (Fig 1C) for transfection. Construction of pKV and pKV-AR1 has been described elsewhere [30]. pKV contained a blasticidin resistance gene (*BSR*) and *d2EGFP* expression cassette. pKV-AR1 (Fig 1C) had a similar structure, but further contained the AR1 sequence from a mouse *Igk* intron that showed strong *in vitro* MAR activity as described above [34]. Construction of pΔBM d2EGFP was conducted as described previously [36], with a nearly identical sequence to that

of pKV in addition to the *DHFR* IR (*Oriβ*; 4,634 bp) harboring a sequence with *in vitro* MAR activity [1].

## Cells, culture, and transfection

The human colorectal carcinoma COLO 320DM cell line was used as for transfection since it can efficiently amplify the IR/MAR plasmid [1]}. COLO 320DM (CCL-220) cells were originally obtained from the American Type Culture Collection (Manassas, VA, USA), and clone "COLO 320DM #3" bearing multiple DMs with an amplified c-*myc* oncogene was obtained by limiting dilution [37]. COLO 320DM cells were cultured in RPMI 1640 medium (Nissui Pharmaceutical Co. Ltd.) supplemented with 10% fetal calf serum (FCS). In addition, the Chinese hamster CHO DG44 cell line (commonly used in industrial recombinant production) was used for evaluation of the practical feasibility of our strategy. CHO DG44 cells were obtained as previously described [8], and cultured in Ham's F-12 medium (Nacalai Tesque Inc.) supplemented with 10% FCS. COLO 320DM cells were transfected using GenePORTER™2 Transfection Reagent (Genlantis Co.), whereas CHO DG44 cells were transfected using Lipofectamine 2000 Reagent (ThermoFisher Scientific Co.) according to the manufacturers' recommended protocols. Stable transformants were selected by blasticidin.

## FISH, flow cytometric analysis and other methods

Metaphase chromosome spreads were prepared according to a standard protocol. A DIG- or biotin-labeled probe was prepared from template DNA using BioPrime DNA Labeling Kit (Invitrogen) with or without 10× DIG DNA Labeling Mixture (Roche), respectively. For flow cytometric analysis to evaluate *d2EGFP* expression, cells were resuspended in phosphate-buffered saline and analyzed using the FACS Calibur system (Becton Dickinson Co.) in the absence or the presence of sodium butyrate for the final 3 days of culturing. The bisulfite treatment followed by the nucleotide sequencing was conducted as previously described [12].

## Results

### The B-3-31 or G5 repeat exhibited long-lasting RIGA

Addition of the B-3-31 or G5 repeat sequence consistently increased the expression of *d2EGFP* from all three vector constructs (pKV, pKV-AR1, and pΔBM.d2EGFP) with or without IR and/or MAR in both the hamster CHO DG44 and human COLO 320DM cells with or without butyrate addition (Fig 2A and 2B). Sodium butyrate, an inhibitor of histone deacetylase complex [38], augmented the expression from the epigenetically silenced repeated sequence [8, 10, 27]. Expression enhancement by B-3-31 or G5 repeat was reproducible among different transfections (see below and S2 Fig) and was statistically significant. Despite only pΔBM.d2EGFP contained the IR/MAR, the repeat of G5, as a core IR, was spontaneously amplified in cells as in the case of the IR/MAR plasmid [30], and the same effect was found for the B-3-31 repeat. Therefore, the expression from all three plasmids was elevated by co-amplification with the B-3-31 or G5 repeat, regardless of the presence of IR/MAR in the plasmid.

This elevated expression persisted for up to 82 days following transfection (Fig 2C), thus indicating the stability of expression enhancement. By contrast, the expression level from the single transfection of pΔBM.d2EGFP or pKV AR1 gradually decreased during this same period (Fig 2C), likely reflecting RIGS of the amplified sequences. Furthermore, co-transfection of pΔBM.d2EGFP and B-3-31 repeat DNA was more effective in elevating gene expression compared to transfection of the pΔBM.d2EGFP vector harboring a cloned B-3-31 sequence

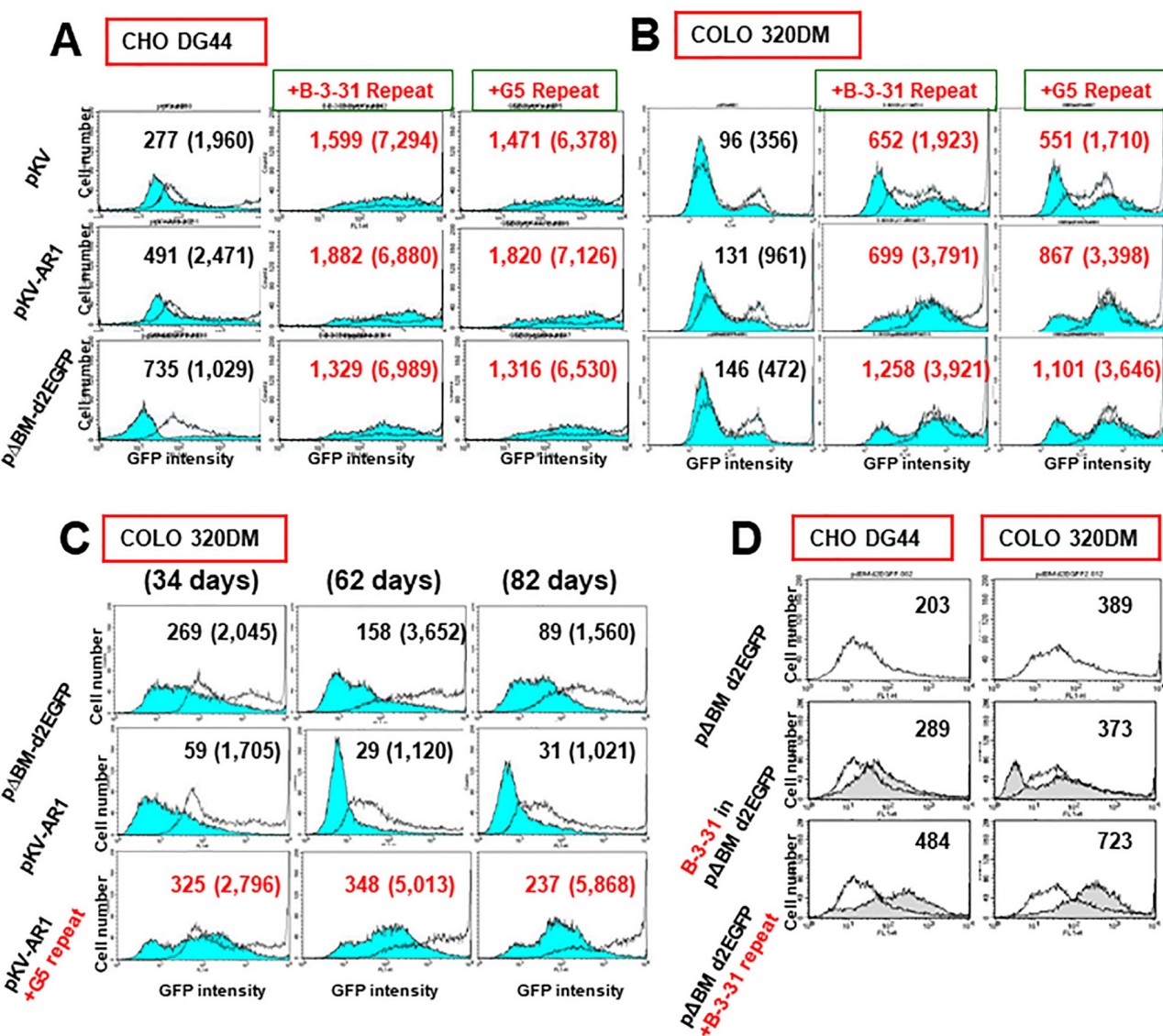

**Fig 2. The B-3-31 repeat or G5 repeat exhibited RIGA.** B-3-31 repeat or G5 repeat DNA was co-transfected with pKV, pKV-AR1 (with a MAR), or pΔBM.d2EGFP (with an IR/MAR) to CHO DG44 cells (A) or COLO 320DM cells (B). The transfected cells were cultured in the presence of 5 μg/ml blasticidin for 39 days (A) or 54 days (B), and analyzed using flow cytometry in the absence (blue-filled lines) or presence (unfilled lines) of 2 mM sodium butyrate during the final 3 days of culturing. The mean fluorescence intensity for the butyrate-absent and butyrate-present (in parentheses) condition is indicated in each graph. (C) The COLO 320DM cells transfected with pΔBM.d2EGFP or pKV-AR1 with or without the G5 repeat were cultured in the presence of blasticidin for 34, 62, and 82 days, and analyzed as described above. (D) pΔBM.d2EGFP, pΔBM.d2EGFP bearing B-3-31 at the AscI site (see Fig 1B), or pΔBM.d2EGFP mixed with B-3-31 repeat DNA was transfected to CHO DG44 cells or COLO 320DM cells, and cultured for 19 days or 36 days, respectively, in the presence of blasticidin. The cells were analyzed using flow cytometry in the absence of butyrate. Unfilled lines show the results for pΔBM.d2EGFP alone and the gray-filled lines show the results for the other two transformants.

(Fig 2D). A similar result was obtained for the G5 repeat in our previous study [30], indicating that enhancement of gene expression by the B-3-31 or G5 repeat was repeat-dependent.

## The core IR or MAR repeats exhibited RIGA

Expression of *d2EGFP* was consistently higher in the culture co-transfected with the core IR from the *c-myc* locus (C12) or *DHFR* locus (D8) or MAR repeat DNA (AR-1 and 32–3), compared with that in the culture transfected with pΔBM.d2EGFP alone (Fig 3A and 3B). This

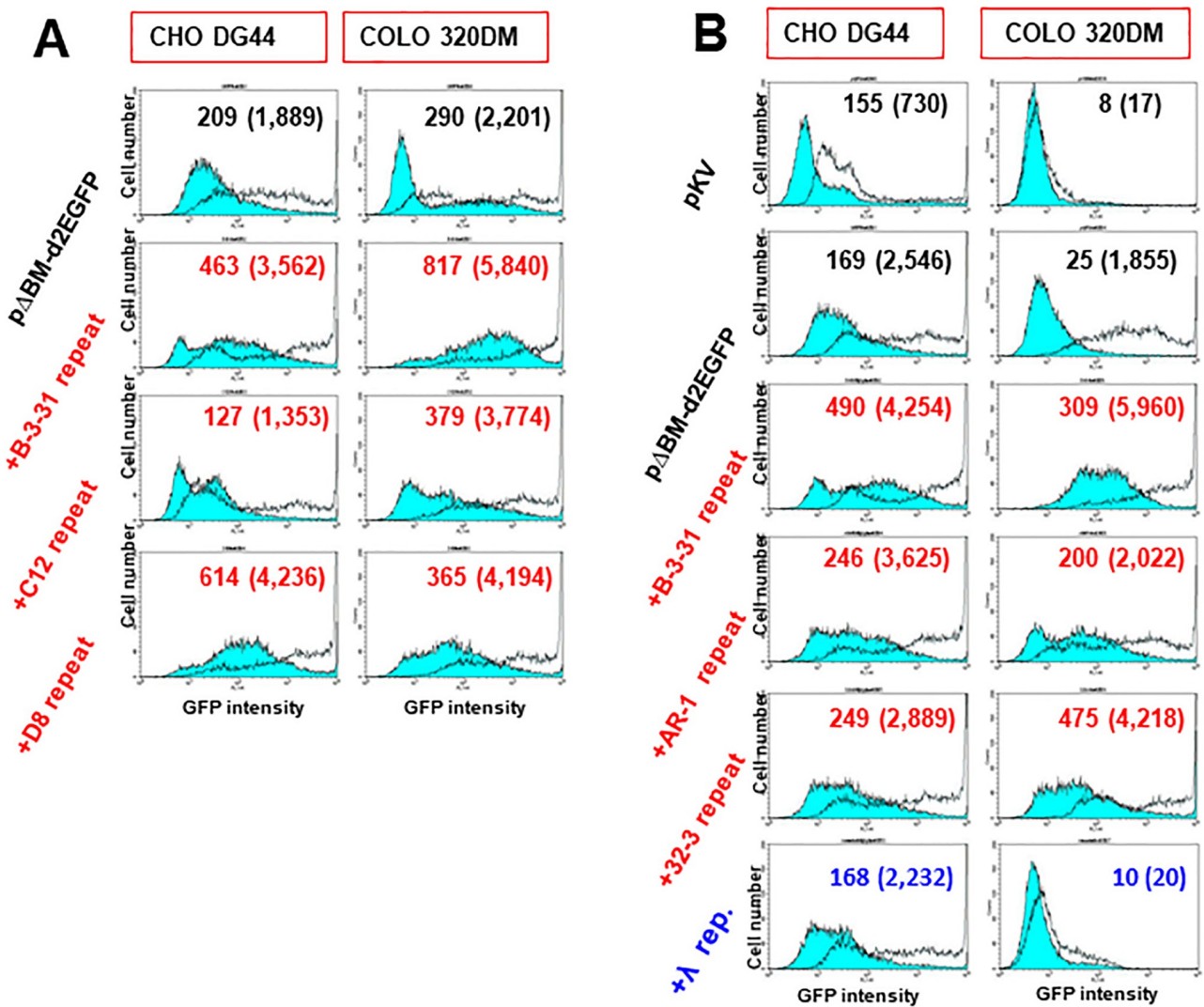

**Fig 3. The core-IR repeat, or MAR repeat exhibited RIGA, whereas the lambda-phage repeat had the opposite effect on gene expression.** CHO DG44 cells or COLO 320DM cells were transfected with pKV, pΔBM.d2EGFP mixed without or with the repeat DNA from B-3-31 as well as C12 or D8 (A), or AR-1, 32–3, or lambda-phage (B). The data shown represent independent transfections. Transfected CHO DG44 cells were cultured in the presence of blasticidin for 31 (A) or 30 (B) days, and COLO 320DM cells were cultured for 56 (A) and 59 (B) days. Flow cytometry results were obtained in the absence (blue-filled lines) or presence (unfilled lines) of 2 mM sodium butyrate during the final 3 days of culturing. The mean fluorescence intensity for the butyrate-absent and butyrate-present (in parentheses) condition is indicated noted in each graph.

effect was observed in both cell lines with and without butyrate addition. Only the C-12 repeat had no significant effect in CHO DG44 cells, whereas it enhanced target gene expression in COLO 320DM cells. Such observation was reproducible among different transfection (S3 Fig). This likely reflects the less efficient amplification of the IR/MAR-containing plasmid in CHO DG44 cells compared to COLO 320DM cells [29].

Therefore, B-3-31, the three core IRs (G5, C12, and D8), and two MARs (AR1 and 32–3) exhibited RIGA. By contrast, under the same experimental conditions (Fig 3B), the repeat of the lambda phage-derived sequence resulted in no apparent change in the expression from amplified pΔBM.d2EGFP in CHO DG44 cells, but rather suppressed the expression to a level lower than that induced by pΔBM.d2EGFP alone in COLO 320DM cells, especially in the butyrate-containing culture. It was considered to reflect RIGS.

## The bacterial sequence and human transposon exhibited RIGS

We next examined the influence of plasmid-derived ColE1, an ampicillin-resistant gene, and three different transposon sequences that are most abundant in the human genome (5′ UTR of LINE, ORF1 of LINE, and Alu). The repeat of all these sequences suppressed gene expression to a level lower than that observed with transfection of pΔBM.d2EGFP alone in both cell lines (Fig 4). This suppressive effect was the most prominent with transfection to COLO 320DM cells and in the presence of butyrate, which was similar to the results obtained for the lambda-phage repeat described above. IR/MAR gene amplification usually proceeds far more efficiently in COLO 320DM cells than in CHO DG44 cells [8, 29], which explains the stronger effect in COLO 320DM cells. The amplified sequences generated from RIGS-inducing sequences in COLO 320DM cells should form tightly compacted chromatin that might not be relaxed by butyrate treatment as a histone deacetylase complex inhibitor. By contrast, the B-3-31 repeat still efficiently enhanced expression of the target gene in the same experiment (Fig 4).

## Statistical significance of RIGS and RIGA

We conducted multiple independent co-transfections of various kinds of plasmids combined with/without B-3-31 or G5 repeats. We then plotted all mean fluorescence intensities values from the flow cytometric analyses as shown in Fig 5. We applied student's t-test to these values and obtained p-values for the expression enhancement by the addition of repeat compared to the plasmid alone (Fig 5A for CHO DG44 and B for COLO 320DM). Results showed that the addition of B-3-31 repeat significantly ($p < 0.05$) elevated the expression from the plasmid in both CHO DG44 and COLO 320DM cell lines. The statistical significance of the addition of G5 repeat was slightly insufficient in CHO DG44 cells, since the number of samples (n) was only 3; however, it was significance in the COLO 320DM cells.

In the above experiments, we have analyzed 6 kinds of sequences that showed RIGA and 6 kinds that showed RIGS. We plotted all mean fluorescence intensities values obtained from the flow cytometric analyses as shown in Fig 5C (CHO DG44) and 5D (COLO 320DM). We then applied student's t-test and obtained p-values for the significant differences between RIGA and RIGS sequences (Fig 5C and 5D). The results showed that, although p-value for butyrate-treated CHO cells was 0.058, the difference was statistically significant ($p < 0.05$) in all other cases.

## CpG content had no effect on gene expression

We previously proposed the "DNA methylation-core and chromatin-spread model" to explain heterochromatin formation by RIGS [12]. Because CpG is methylated in mammalian cells, we determined the CpG content among the sequences examined in this study. Almost all sequences exhibiting RIGA had a low CpG content, whereas most sequences exhibiting RIGS had a high CpG content, except for C12 and ORF1 of LINE (Fig 6A). C12 is located at the c-*myc* promoter region, and thus might represent a CpG island. ORF1 of LINE had a lower CpG content among the RIGS sequences, but it remained much higher than that of the RIGA sequence. This clear difference of CpG content between RIGS and RIGA sequences suggested that the high CpG content in RIGS sequences might initiate repeat heterochromatinization. To validate this, we treated the lambda-phage DNA with bisulfite to convert unmethylated cytosine to uracil, which was then cloned in *Escherichia coli* cells. The sequenced result (Fig 6B) revealed the successful conversion from C to T, which diminished the CpG content and consequently increased the AT content. Compared with the untreated sequence, bisulfite treatment of the lambda-phage sequence had no effect on the gene expression from pΔBM. d2EGFP, both when the repeat DNA was co-transfected with pΔBM.d2EGFP (Fig 6C) and

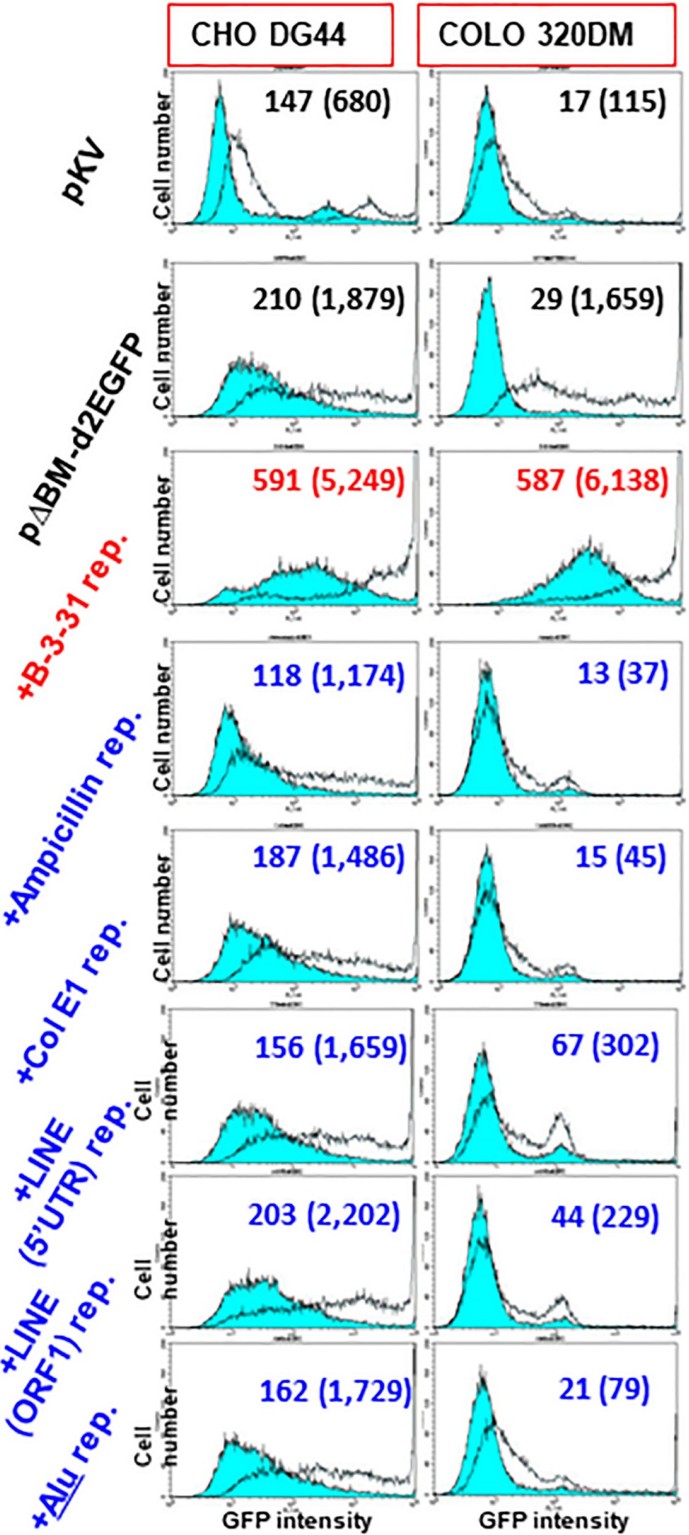

**Fig 4. The plasmid or transposon repeat exhibited RIGS.** (A) CHO DG44 cells or COLO 320DM cells were transfected with pKV, pΔBM.d2EGFP mixed with or without the repeat of B-3-31, plasmid-derived ampicillin resistance gene, Colicin E1, 5′ UTR or ORF1 of human LINE, and human Alu. The stable transformants were selected by blasticidin for 23 days for CHO DG44 cells or for 37 days for COLO 320DM cells. Flow cytometry results were obtained in the absence (blue-filled lines) or presence (unfilled lines) of 2 mM sodium butyrate during the last 3 days. The mean fluorescence intensity for the butyrate-absent and butyrate-present (in parentheses) condition is noted in each graph.

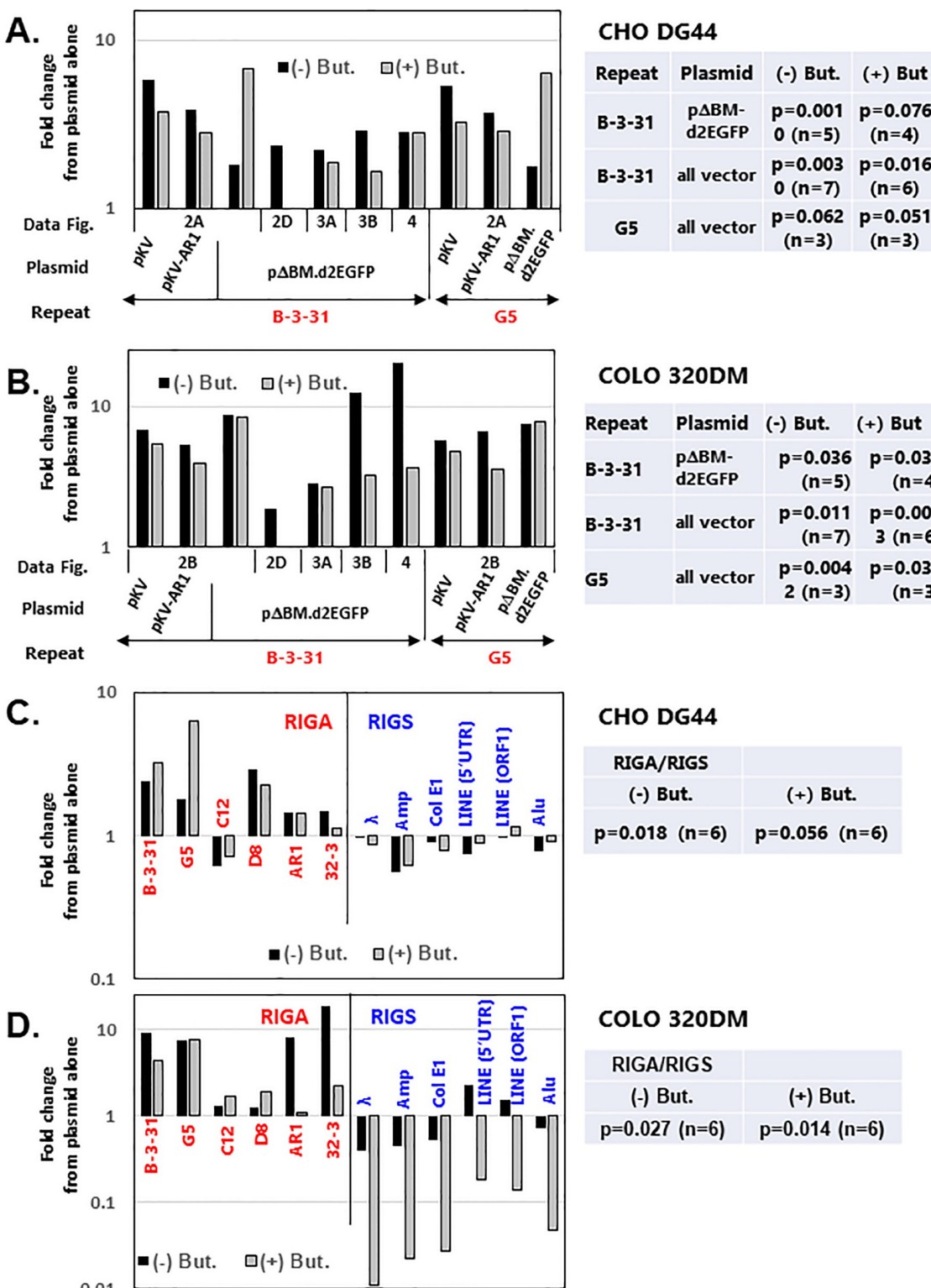

**Fig 5. Statistical significance of RIGS and RIGA.** The mean fluorescence intensities of flow cytometric analyses, shown in Figs 2–4, were plotted in graphs. The p-values for the expression enhancement by the addition of B-3-31 or G5 repeat compared to the plasmid alone were calculated and shown in the tables, for CHO DG44 cells (A) and for COLO 320DM cells (B). The p-values for the difference between the effects of 6 different RIGA and 6 different RIGS sequences were calculated and shown in the tables, for CHO DG44 cells (C) and for COLO 320DM cells (D).

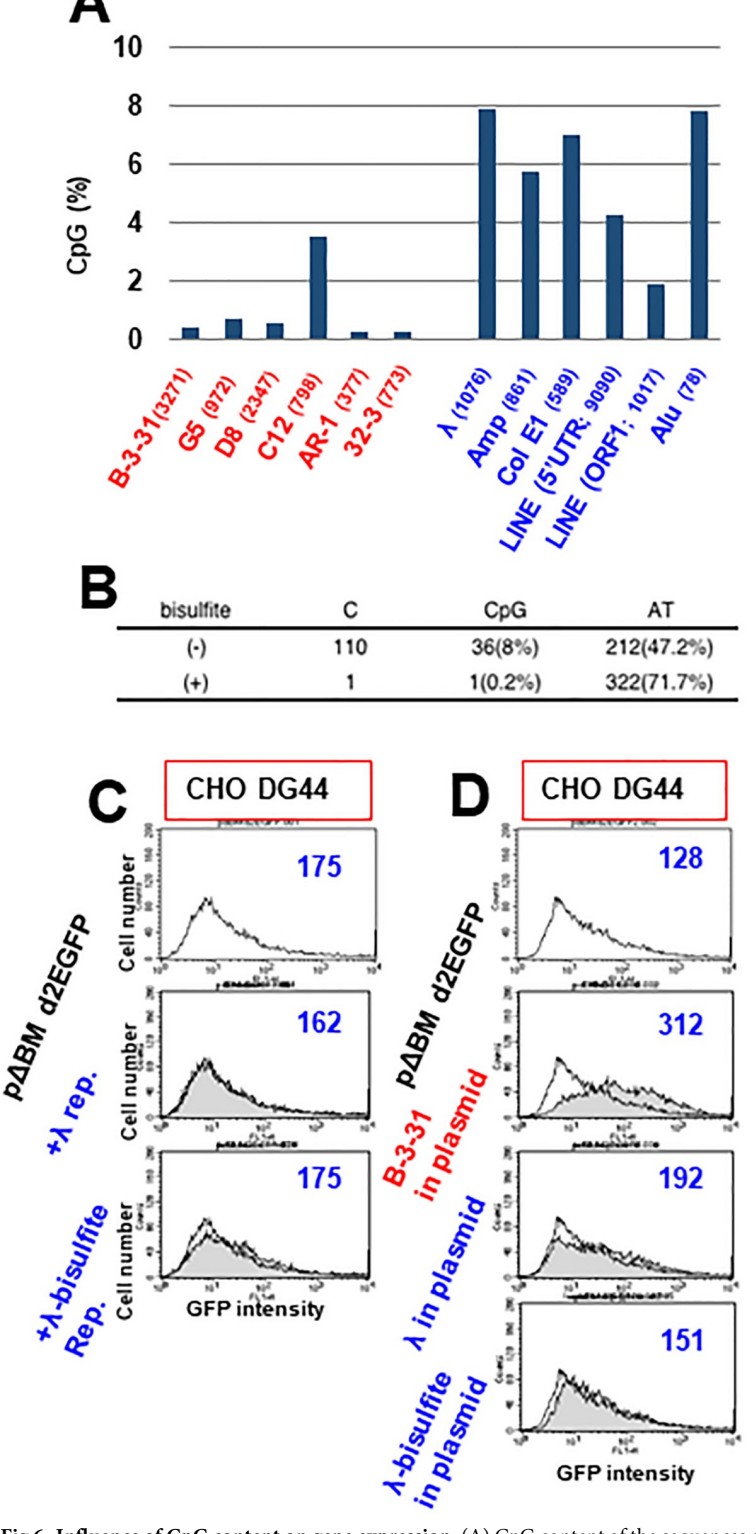

**Fig 6. Influence of CpG content on gene expression.** (A) CpG content of the sequences examined in this study. (B) Lambda-phage DNA was treated with bisulfite to convert unmethylated cytosine to uracil, followed by PCR amplification and cloning in *E. coli* host cells. The extent of conversion was determined by sequencing. (C) The repeat DNA of untreated or bisulfite-treated sequences was mixed with pΔBM.d2EGFP and co-transfected to CHO DG44 cells. (D) The B-3-31, untreated or bisulfite-treated lambda-phage sequence was cloned in pΔBM.d2EGFP and

transfected to CHO DG44 cells. Cells were cultured in medium containing blasticidin for 24 (C) or 52 (D) days, and *d2EGFP* expression was analyzed using flow cytometry in the absence of butyrate. Unfilled lines indicate pΔBM. d2EGFP single transformants and gray-filled lines indicate the test transformants.

when the sequence was cloned in pΔBM.d2EGFP (Fig 6D). However, in the same experiment, the B-3-31 repeat significantly enhanced gene expression.

## Autonomous replication initiation might be related to RIGA

During this study, we found that the transformation efficiency, as judged by the primary colony number, was significantly higher with transfection of the RIGA repeats (Fig 7A). A high

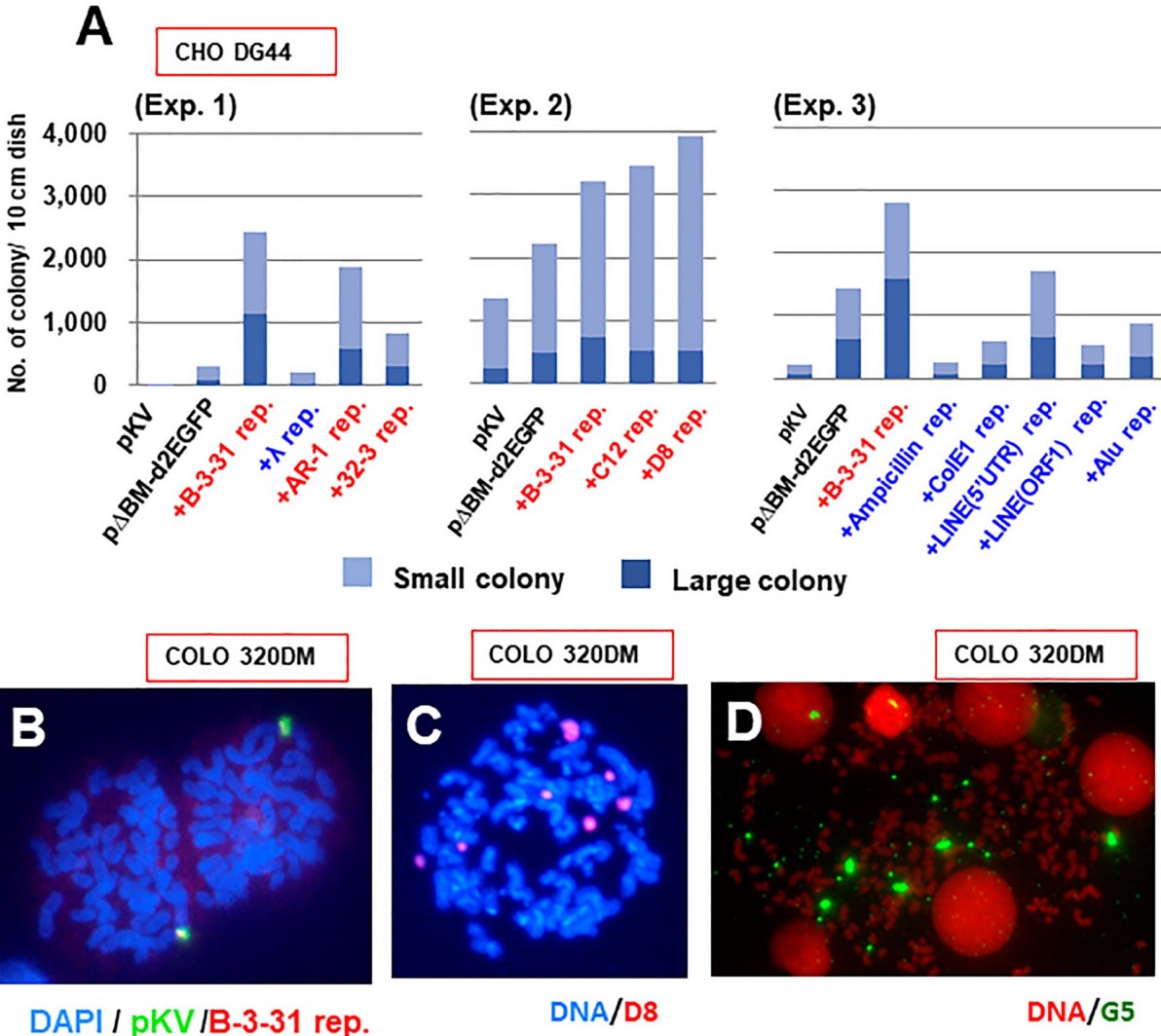

**Fig 7. RIGA repeats might support extrachromosomal replication initiation.** (A) Thirteen (Exp. 1, 3) or 22 (Exp. 2) days following transfection of the indicated DNA to CHO DG44 cells, the colony number per dish was counted and plotted; a small and large colony was defined as a colony comprising less than or more than 100 cells, respectively. (B) The B-3-31 repeat was co-transfected with pKV to COLO 320DM cells. The metaphase spread was prepared from the stable transformants selected by blasticidin, and simultaneously hybridized with the DIG-labeled pKV probe and biotin-labeled B-3-31 probe, which was detected using green and red fluorescence, respectively. The D8 repeat (C) or G5 repeat (D) was co-transfected with pΔBM.d2EGFP to COLO 320DM cells. The metaphases spread was hybridized with a probe prepared from the D8 repeat DNA (B) or G5 repeat DNA (C) and visualized as green fluorescence. The DNA was counterstained in blue by 4',6-diamidino-2-phenylindole (B, C) or in red by propidium iodide (D).

transformation efficiency is usually considered to reflect episomal replication of the transfected DNA, and the autonomous replication sequence was cloned in yeast cells based on the increase in the transformation efficiency [39]. Furthermore, we routinely observed a higher transformation efficiency from the IR/MAR plasmid compared to the normal plasmid [30], which explains the higher colony number of pΔBM.d2EGFP than pKV (Fig 7A). Such enhancement by IR/MAR was more evident in the number of small colonies comprising less than 100 cells. Such small colony is thought to reflect the transient episomal maintenance during the initial few days following the transfection. We found that the transfection of the RIGA repeat produced higher colony number, especially for the small colony, while RIGS repeat did not (Fig 7A).

Most transformed cells produced by co-transfection of pKV and the B-3-31 repeat harbored the co-amplified plasmid and B-3-31 sequences (Fig 7B). Since the transfection of pKV alone never generated amplified structure, as it is a normal plasmid, this suggest that the B-3-31 repeat can be spontaneously amplified, as in the case of the core IR repeat sequences [30]. We also observed many unusually giant extrachromosomal signals in cells co-transfected with the D8 or G5 repeat (Fig 7C and 7D), which might have been generated from replication anomalies caused by the repeat. These findings suggest that autonomous replication initiation of the repeated sequence might be related to the RIGA phenomenon.

### RIGA repeats favored chromosomal amplification

FISH analysis of the metaphase spread from cells using the probe prepared from pΔBM. d2EGFP DNA showed that the IR/MAR-bearing pΔBM.d2EGFP plasmid was amplified in COLO 320DM cells both at the HSR and the DMs (Fig 8), in line with previous results [4]. Surprisingly, there was a strong tendency for co-transfection with the RIGA repeats to cause preferential amplification at the chromosomal HSR (Fig 8).

## Discussion

We applied our novel assay system (Fig 1A) based on *in vitro* ligation followed by intracellular amplification with our original IR/MAR technology to examine the roles of sequence repeats in silencing gene expression. A similar strategy was used in our previous study [30], except here, only used a direct repeat for transfection since the inverted repeat was found to be structurally unstable [30]. With this assay system, the tested sequences could be clearly classified into two main categories: sequences that exhibit RIGS and those that exhibit RIGA. The RIGA phenomenon suggested that the sequence repeat *per se* does not necessarily silence gene expression but could even upregulate gene expression depending on the specific repeated sequence. This finding helps us to understand the nature and roles of repeat sequences in the mammalian genome, along with direct application value for recombinant production. However, elucidating the underlying mechanism of RIGA or RIGS remains warranted.

One apparent difference between the sequences exhibiting RIGS or RIGA was the CpG and/or AT content. We have previously reported that the 3271-bp B-3-31 sequence did not contain the core region or the non-B structure that typically contribute to gene expression activation [27]. Therefore, we suggested that a low CpG content or high AT content throughout B-3-31 might activate the gene if in proximity to the promoter. However, the present findings demonstrated that bisulfite treatment, which diminished the CpG content and increased the AT content, of the lambda-phage sequence did not influence gene expression, suggesting that even if the CpG/AT content is involved in regulating gene expression, it is not the single determinant. We have found that most repeated sequences generated by the IR/MAR plasmid were CpG hypomethylated, and only few sequences among the repeats were hypermethylated [12].

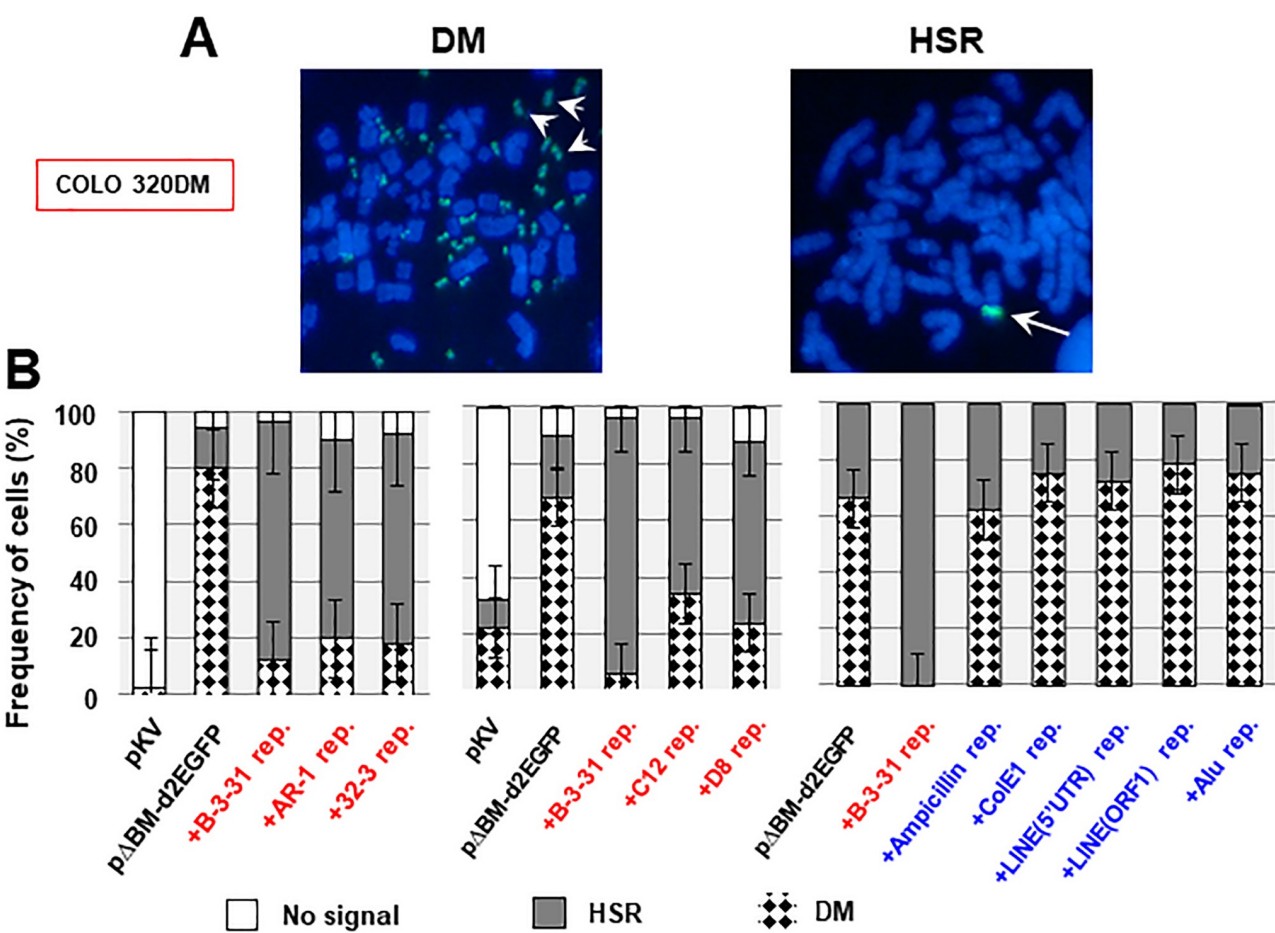

**Fig 8. The RIGS repeat favored extrachromosomal amplification, whereas the RIGA repeat favored chromosomal amplification.** The metaphase spread from the indicated transfection to COLO 320DM cells was subjected to FISH using the probe prepared from pΔBM.d2EGFP DNA. (A) Representative images for the amplification at DMs (arrow heads) or HSR (arrow). (B) Frequency of cells showing amplification of either DMs or HSR, which was counted by examining more than 50 metaphase cells, three times each; data are presented as the mean ± standard deviation.

This was consistent with hypomethylation of CpG at centromeric repeats [40] or transgene arrays [25], both of which were heterochromatinized by RIGS. A recent study showed that integration of CpG-free DNA induced the *de novo* methylation of CpG islands [41]. Therefore, the hypomethylated state of the repeat might contribute to the hypermethylation of limited sequences by an unknown mechanism, which in turn could initiate heterochromatinization of the entire repeat.

Another apparent feature of the sequences exhibiting RIGA was their influence on replication initiation. We found that the RIGA repeat enhanced the transformation efficiency, generated extraordinarily large extrachromosomal signals, and reduced extrachromosomal gene amplification. This last effect is likely related to our previous finding that double-strand breakage in multiple DMs resulted in their aggregation and elimination from cells, or morphological transformation to the HSR [32, 42]. Thus, a higher level of transcription and/or higher frequency of replication initiation at the RIGA repeat might induce double strand breaks in the amplicon, which could eliminate the DMs or convert them into HSR. Indeed, the core IRs (G5, C12, and D8) should be tightly associated with replication initiation, and the G5 repeat was quite efficiently amplified in cells [30]. Nuclear matrix attachment (i.e., MARs) was

suggested to be involved in DNA replication initiation [43, 44]. The B-3-31 repeat initiated gene amplification, whereas the IR (considered the "replicator" of DNA replication or the replication origin) was reported to enhance gene expression [31]. Many reports have demonstrated that the MAR enhanced transgene expression [45–47]. Therefore, there is an association between sequences that support replication initiation, sequences that augment gene expression, and the RIGA phenomenon.

Transgene expression in animal cells is typically achieved using a plasmid vector and *E. coli* host. Herein, we showed that the Amp gene or ColE1 sequence in a plasmid vector exhibited RIGS. This finding is consistent with the ability of a plasmid bacterial sequence to silence cis-ligated transgene expression in animal cells [48], which was subsequently shown to occur in a CpG-independent manner [49]. Therefore, to improve transgene expression, especially using a gene amplification strategy, sequences that exhibit RIGS should be avoided, and those exhibiting RIGA should be preferentially incorporated.

## Supporting information

**S1 Fig. PCR primers used in this study.**
(PPTX)

**S2 Fig. Experiment same as Fig 2B, but different transfection.**
(PPTX)

**S3 Fig. Experiment same as Fig 3A (CHO DG44), but different transfection.**
(PPTX)

**S1 Raw images.**
(PDF)

## Acknowledgments

We would like to Dr. Ken Tsutsui of Okayama University for kindly gifting the MAR sequences. We would also like to thank the Hiroshima University Natural Science Center for Basic Research and Development for use of the flow cytometer.

## Author Contributions

**Conceptualization:** Noriaki Shimizu.

**Data curation:** Yusuke Ogaki, Noriaki Shimizu.

**Formal analysis:** Yusuke Ogaki, Miki Fukuma, Noriaki Shimizu.

**Funding acquisition:** Noriaki Shimizu.

**Investigation:** Yusuke Ogaki, Miki Fukuma, Noriaki Shimizu.

**Methodology:** Yusuke Ogaki, Miki Fukuma, Noriaki Shimizu.

**Project administration:** Noriaki Shimizu.

**Resources:** Noriaki Shimizu.

**Software:** Noriaki Shimizu.

**Supervision:** Noriaki Shimizu.

**Validation:** Noriaki Shimizu.

**Visualization:** Noriaki Shimizu.

**Writing – original draft:** Yusuke Ogaki, Miki Fukuma, Noriaki Shimizu.

**Writing – review & editing:** Noriaki Shimizu.

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
