## [Decision Letter · Decision Letter 0]

27 Apr 2020

PONE-D-20-07143

Repeat induces not only gene silencing (RIGS), but also gene activation (RIGA) in mammalian cells

PLOS ONE

Dear Prof. Shimizu,

Thank you for submitting your manuscript to PLOS ONE. After careful consideration, we feel that it has merit but does not fully meet PLOS ONE’s publication criteria as it currently stands. Therefore, we invite you to submit a revised version of the manuscript that addresses the points raised during the review process.

The following set of changes are required  to address the Reviewers' and AE's comments regarding textual clarity, statistical analysis and scientific rigor:

1. Textual Issues

First, the manuscript should be proofread by another investigator familiar with journal publication, to eliminate grammatical errors that are present throughout.  Also please place line numbers in the text for easier textual reviewing.Second, the manuscript is written for a specialty audience.A) The information in the abstract and the manuscript should be accessible to a general audience, eliminating excess technical jargon.B) The introduction should provide more background regarding RIGS, a less technical description of the labs’ previous work, and an elimination of methods/results.  C) The Methods should be describe more fully and include statistical analysis.D) The Results must provide more introductory context (e.g, reason for treatment with butyrate), hypothesis to be tested, and a better textual description of the data presented in the Figures, and a simple conclusion from the data shown.E) In the Discussion, the following statement needs to be explained more fully: “Transgene expression in animal cells is typically achieved using a plasmid vector and E. coli host. Here, we showed that the Amp gene or ColE1 sequence exhibited RIGA. This finding is consistent with the ability of a plasmid bacterial sequence to silence episomal transgene expression [39}.F) The Figure Legends must better describe the data presented, the lane designation, abbreviations, and colors. 

2. Statistical Evaluation: 

 Insufficient information is presented including the number of experimental repeats, the statistical tests used, and the statistical significance of any increase or decrease in gene expression (including the modifications suggested Reviewer 2).

3. Scientific Rigor

As indicated by Reviewer 1, corrections based on the differing transfection efficiencies must be presented and discussed.Provide the data requested on the number of repeats units introduced after transfection,Provide the clarification of Figures 6 and 7 explained by Reviewer 2. Provide answers to all additional issues raised by the Reviewers. Provide all raw data or links to the raw data as given in the Author guidelines.

The first and second reviewers provided non-overlapping but sound critiques.

The AE's evaluation concurred with Reviewers critiques, but also identified required improvements in the text and the determination of statistical significance (as described above) 

We would appreciate receiving your revised manuscript by Jun 11 2020 11:59PM. To enhance the reproducibility of your results, we recommend that if applicable you deposit your laboratory protocols in protocols.io, where a protocol can be assigned its own identifier (DOI) such that it can be cited independently in the future. For instructions see: http://journals.plos.org/plosone/s/submission-guidelines#loc-laboratory-protocols

We look forward to receiving your revised manuscript.

Kind regards,

Arthur J. Lustig, PhD

Academic Editor

PLOS ONE

Journal Requirements:

Reviewers' comments:

Reviewer's Responses to Questions

**Comments to the Author**

1. Is the manuscript technically sound, and do the data support the conclusions?

Reviewer #1: Partly

Reviewer #2: Partly

2. Has the statistical analysis been performed appropriately and rigorously? 

Reviewer #1: No

Reviewer #2: No

3. Have the authors made all data underlying the findings in their manuscript fully available?

Reviewer #1: Yes

Reviewer #2: Yes

4. Is the manuscript presented in an intelligible fashion and written in standard English?

Reviewer #1: Yes

Reviewer #2: Yes

5. Review Comments to the Author

Reviewer #1: The manuscript presents an interesting set of observations and indicates that certain sequences, when repeated, may induce silencing (RIGS) or activate gene expression (RIGA). This type of phenomenon have been known from past studies, however, this work associates gene expression activation with potential replicative properties of the sequences.

Results, although interesting, are not supported by appropriate controls and therefore I found some of the conclusions made by the authors over interpretation of the presented data.

First, experiments are based on transfection and a careful control of transfection efficiency should be presented. In addition, it is not clear how many copies of the initially transfected plasmids are in fact integrated into the genome and how many (if any are in the episomal state). This could be determined by appropriate qPCR approaches.

Finally, all expression determinations are made with a GFP encoding construct under CMV promoter. If this determinations of RIGS and RIGA could address potentially efficiency of recombinant protein production in the cells, a set of different promoters frequently used in mammalian expression systems should be tested to demonstrate universality of the observations.

Minor issues:

The manuscript is well written and presented, however, parts of the introduction are in fact description of the methods/results (page). This should be replace with more general presentation and importance of the subject.

Reviewer #2: In the MS entitled “Repeat induces not only gene silencing (RIGS), but also gene activation (RIGA) in mammalian cells” Ogaki developed an assay system for evaluating the influence of repeat sequences on gene expression, based on in vitro ligation followed by the previously published IR (Initiation Region)/ MAR (Matrix Attachment Region) gene amplification technology in mammalian cell. This technology is use to generate high protein expression in mammalian cells. The novel contribution of the study is the analysis of human genomic repeats (e.g. LINEs, Alus), some other no specified repeats found in the genome named (B-3-31) that enhanced gene expression from tandemly amplified repeats and some core sequences from different initiation regions that the authors called C12, D8, and G5. The Shimizu lab has a great deal of experience using the IR/MAR technology applied in this study. The author’s results, therefore, highlights the potential of repeat sequences may downregulate (RIGS) or upregulate (RIGA) the expression of eGFP using the IR/MAR system. The manuscript is well written and it fits well with the scope of the journal. After the revisions have been done, I recommend publishing this article.

I have read this paper several times and submit these recommendations to you:

Results:

It will be helpful if Figure 1D it is described in more detail in the figure legend. Please add sizes of the band for each gel and define the abbreviation M and L in the figure legend.

For Figure 2, add a graph that summarizes the % eGFP cells and the fold effects of significant differences using the mean. The way the data is shown and the description in the results and figure legend implies that the experiments were done only one time (n=1). Do this for A, B, C, D. Please add statistical analysis to the graphs.

Figure 3 and 4 do the same as described for Figure 2.

For Figure 5C and 5D do the same as described for Figure 2.

For Figure 6A the figure legend describes “the colony number per dish was counted and plotted; a small and large colony was defined as a colony comprising less than or more than 100 cells, respectively”. The manuscript does not explain why the size is important. Please explain why the data was presented this way. Is there an actual physical measure you used to determine 100 cells? If so, please add to the figure legend.

Figure 7B 3rd graph showing RIGS looks the same as baseline pBM-d2EGFP. Please shoe the % of the DMs. This data does not support that RIGS result in “preferential amplification at the extrachromosomal DMs” as stated in the results. Especially when you compared the baseline pBM-d2EGFP from the RIGA experiments.

6. PLOS authors have the option to publish the peer review history of their article (what does this mean?). If published, this will include your full peer review and any attached files.

Reviewer #1: No

Reviewer #2: No

---

## [Author Response · Author response to Decision Letter 0]

19 May 2020

We wrote all the responses to editor and reviewers in "response to reviewers".

---

## [Decision Letter · Decision Letter 1]

10 Jun 2020

Repeat induces not only gene silencing, but also gene activation in mammalian cells

PONE-D-20-07143R1

Dear Dr. Shimizu,

We’re pleased to inform you that your manuscript has been judged scientifically suitable for publication and will be formally accepted for publication once it meets all outstanding technical requirements.

Kind regards,

Arthur J. Lustig, PhD

Academic Editor

PLOS ONE

Additional Editor Comments (optional):

Reviewers' comments:

Reviewer's Responses to Questions

**Comments to the Author**

1. If the authors have adequately addressed your comments raised in a previous round of review and you feel that this manuscript is now acceptable for publication, you may indicate that here to bypass the “Comments to the Author” section, enter your conflict of interest statement in the “Confidential to Editor” section, and submit your "Accept" recommendation.

Reviewer #1: All comments have been addressed

Reviewer #2: All comments have been addressed

2. Is the manuscript technically sound, and do the data support the conclusions?

Reviewer #1: Yes

Reviewer #2: Yes

3. Has the statistical analysis been performed appropriately and rigorously? 

Reviewer #1: Yes

Reviewer #2: No

4. Have the authors made all data underlying the findings in their manuscript fully available?

Reviewer #1: Yes

Reviewer #2: Yes

5. Is the manuscript presented in an intelligible fashion and written in standard English?

Reviewer #1: Yes

Reviewer #2: Yes

6. Review Comments to the Author

Reviewer #1: The Authors made significant improvements in the manuscript. All reviewer comments and criticisms were addressed by appropriate changes in the text of explanation/refences to their previous work.

Reviewer #2: (No Response)

7. PLOS authors have the option to publish the peer review history of their article (what does this mean?). If published, this will include your full peer review and any attached files.

Reviewer #1: No

Reviewer #2: No

---

## [Editor Report · Acceptance letter]

15 Jun 2020

PONE-D-20-07143R1 

Repeat induces not only gene silencing, but also gene activation in mammalian cells 

Dear Dr. Shimizu:

I'm pleased to inform you that your manuscript has been deemed suitable for publication in PLOS ONE. Congratulations! Your manuscript is now with our production department. 

Kind regards, 

on behalf of

Dr. Arthur J. Lustig 

Academic Editor

PLOS ONE